# 3q26.2/*MECOM* Rearrangements by Pericentric Inv(3): Diagnostic Challenges and Clinicopathologic Features

**DOI:** 10.3390/cancers15020458

**Published:** 2023-01-11

**Authors:** Zhenya Tang, Wei Wang, Su Yang, Hanadi El Achi, Hong Fang, Karen Amelia Nahmod, Gokce A. Toruner, Jie Xu, Beenu Thakral, Edward Ayoub, Ghayas C. Issa, C. Cameron Yin, M. James You, Roberto N. Miranda, Joseph D. Khoury, L. Jeffrey Medeiros, Guilin Tang

**Affiliations:** 1Department of Hematopathology, The University of Texas MD Anderson Cancer Center, Houston, TX 77030, USA; 2Department of Leukemia, The University of Texas MD Anderson Cancer Center, Houston, TX 77030, USA

**Keywords:** *MECOM* rearrangement, pericentric inversion, −7/7q-, immunophenotype, mutation profile

## Abstract

**Simple Summary:**

In this study, 17 acute myeloid leukemia (AML) patients were identified to pose a pericentric inv(3) aberration with breakpoints at 3p23 (n = 11), 3p25 (n = 3), 3p21 (n = 2) and 3p13 (n = 1) on 3p and 3q26.2 on 3q, leading to *MECOM* rearrangement (*MECOM-R*). These pericentric inv(3)s were overlooked by karyotyping initially in 16 of 17 cases and later detected by metaphase FISH analysis. Compared to AML patients with classic/paracentric inv(3)(q21q26.2), our patients with pericentric inv(3)s also exhibited frequent cytopenia, morphological dysplasia (especially megakaryocytes), −7/del(7q) and dismal outcomes (median overall survival: 14 months). However, the patients in this cohort also exhibited certain unique features: high frequencies of thrombocytopenia (n = 15, 88%) and monocytosis in peripheral blood (n = 15, 88%) and decreased megakaryocytes (n = 11, 65%). In summary, the pericentric inv(3)s are often subtle/cryptic by chromosomal analysis. A reflex FISH analysis for *MECOM-R* is recommended in myeloid neoplasms showing −7/del(7q).

**Abstract:**

*MECOM* rearrangement (*MECOM-R*) resulting from 3q26.2 aberrations is often associated with myeloid neoplasms and inferior prognosis in affected patients. Uncommonly, certain 3q26.2/*MECOM-R* can be subtle/cryptic and consequently overlooked by karyotyping. We identified 17 acute myeloid leukemia (AML) patients (male/female: 13/4 with a median age of 67 years, range 42 to 85 years) with a pericentric inv(3) leading to *MECOM-R,* with breakpoints at 3p23 (n = 11), 3p25 (n = 3), 3p21 (n = 2) and 3p13 (n = 1) on 3p and 3q26.2 on 3q. These pericentric inv(3)s were overlooked by karyotyping initially in 16 of 17 cases and later detected by metaphase FISH analysis. Similar to the patients with classic/paracentric inv(3)(q21q26.2), patients with pericentric inv(3) exhibited frequent cytopenia, morphological dysplasia (especially megakaryocytes), −7/del(7q), frequent *NRAS* (n = 6), *RUNX1* (n = 5) and *FLT-3* (n = 4) mutations and dismal outcomes (median overall survival: 14 months). However, patients with pericentric inv(3) more frequently had AML with thrombocytopenia (n = 15, 88%), relative monocytosis in peripheral blood (n = 15, 88%), decreased megakaryocytes (n = 11, 65%), and lower *SF3B1* mutation. We conclude that AML with pericentric inv(3) shares some similarities with AML associated with classic/paracentric inv(3)/*GATA2::MECOM* but also shows certain unique features. Pericentric inv(3)s are often subtle/cryptic by chromosomal analysis. A reflex FISH analysis for *MECOM-R* is recommended in myeloid neoplasms showing −7/del(7q).

## 1. Introduction

Approximately 4% of patients with acute myeloid leukemia (AML) have neoplasms associated with rearrangements of the myelodysplasia syndrome 1 (*MDS1*) and ecotropic viral integration site 1 (*EVI1*) complex locus (*MECOM*). *MECOM* rearrangement (*MECOM*-*R*) is a biomarker for disease progression and inferior prognosis in AML patients [1,2,3,4,5] and is considered as a defining genetic abnormality for AML regardless of blasts count under the 5th World Health Organization Classification for hematologic malignancies [6]. In 30–50% of these cases, *MECOM*-*R* is derived from the classic inv(3)(q21q26.2)/t(3;3)(q21;q26.2) or inv(3)/t(3;3), in which the super-enhancer of GATA binding protein 2 (*GATA2*) located at 3q21 has been hijacked by *MECOM* located at 3q26.2, resulting in a cascade of transcriptional dysregulation including the inhibition of *GATA2* expression and the overexpression of *MECOM* [1,7,8]. In the remaining AML cases, *MECOM*-*R* can be caused by a wide spectrum (over 120 types) of chromosomal aberrations involving 3q26.2 [9,10]. The term “atypical 3q26.2/*MECOM-R*” has been used to describe this subgroup of cases [11]. In some cases with atypical 3q26.2/*MECOM-R*, chimeric transcripts have been identified, such as t(2;3)(p21;q26.2) with *THADA::MECOM* fusion [12,13], t(3;7)(q26.2;q21) with *CDK6::MECOM* fusion [11,14,15,16,17], t(3;12)(q26.2;p13) with *ETV6::MECOM* fusion [18,19,20,21,22,23,24,25,26,27,28,29,30,31,32,33,34], t(3;21)(q26.2;q11.2) with *NRIP1::MECOM* fusion [16,35,36], t(3;21)(q26.2;q22) with *RUNX1::MECOM* fusion [37,38,39,40] and so on. However, no chimeric transcript has been isolated in most AML cases with atypical 3q26.2/*MECOM-R*; therefore, the underlying mechanism of dysregulation of *MECOM* in these cases very likely mimics cases with typical 3q26.2/*MECOM-R*, as mentioned above. For example, in a group of AML cases with t(3;8)(q26.2;q24.2)/*MECOM-R* [41,42,43,44], *MECOM* and *MYC* are not re-joined to each other. Instead, a super-enhancer-located downstream of *MYC* (MYCSE) has been translocated upstream of the promotor of *MECOM*, which consequently upregulates *MECOM* expression, as demonstrated in both clinical cases and engineered cell models [44].

Up to 15% of AML cases have high *MECOM* expression, as detected by quantitative reverse transcription polymerase chain reaction (RT-PCR), almost triple the percentage of AML cases with 3q26.2/*MECOM-R* detected by conventional cytogenetics [2,3,45,46,47]. There are multiple factors associated with this phenomenon. First, certain AML cases may have *MECOM* overexpression but without 3q26.2/*MECOM-R*. For example, in a study of transcriptome profiles in pediatric AML by Shiba et al. [30], 30% (9/30) of AML cases with t(9;11)(p21;q23)/*KMT2A::MLLT3*, representing 50% of all *KMT2A (MLL)* rearrangement (*KMT2A-R*) AML cases in their cohort, and over 50% of cases with acute megakaryoblastic leukemia (AML with FAB-M7) overexpress *MECOM* but without 3q26.2/*MECOM-R*. The mechanisms related to *MECOM* overexpression remain unknown in these cases. Second, some chromosomal aberrations related to *MECOM-R* may be complex and are often categorized as unidentifiable marker chromosome(s). An assessment by a fluorescence in situ hybridization (FISH) assay is often necessary and beneficial for identifying the *MECOM-R* status in these cases [9,10]. Third, some AML cases may possess *MECOM* amplification instead of *MECOM-R* as a cause of *MECOM* overexpression. A segmental duplication/amplification involving 3q26.2/*MECOM* manifesting as an intra-chromosomal homogenous staining region (HSR) or extra-chromosomal double minutes (dm) may occur in these cases [48,49,50]. Lastly, certain chromosomal aberrations related to *MECOM-R* can be subtle or even cryptic when studied by conventional cytogenetics. For example, t(3;21)(q26.2;q11)/*NRIP1::MECOM*, t(3;7)(q26.2;q21)/*CDK6::MECOM* and inv(3)(p24q26)/*MECOM-R* have been considered as cryptic for chromosomal analysis. They could be easily overlooked by chromosomal analysis only. FISH testing, especially metaphase FISH or, ideally, mapback FISH testing performing on previously G-banding analyzed metaphase(s), is usually required to identify/confirm *MECOM-R* and the underlying subtle/cryptic aberrations [16].

In this study, we reported a cohort of 17 myeloid neoplasms associated with *MECOM-R* via pericentric inv(3)s at various band levels of 3p. These abnormalities were mostly overlooked by chromosomal analyses initially. This subset of 3q26.2/*MECOM-R* cases arising via pericentric inv(3) is associated with some characterized pathological features when compared with *MECOM-R* cases that arise through the classic inv(3)(q21q26.2)/t(3;3)(q21;q26.2).

## 2. Materials and Methods

### 2.1. Case Selection

We searched the cytogenetics database at The University of Texas MD Anderson Cancer Center (MDACC) for cases with pericentric inv(3) involving 3q26.2/*MECOM-R*, as confirmed by FISH results from 1 May 2009 to 30 September 2022. The clinicopathologic and other laboratory information of selected cases was collected through electronic medical record review. This study was approved by the MDACC Institutional Review Board (IRB) and conducted in accordance with the Declaration of Helsinki.

### 2.2. Conventional Karyotype Analysis

Conventional cytogenetic analysis (G-banded chromosomal analysis or karyotyping) was performed on unstimulated 24 h and 48 h bone marrow aspirate cultures using standard techniques. A total of 20 metaphases were routinely analyzed by two technologists, and the results were reported following the 2020 International System for Human Cytogenetics Nomenclature (ISCN 2020) guidelines [10,43]. A complex karyotype is defined as ≥3 clonal chromosomal abnormalities, of which at least one structural chromosomal abnormality is present.

### 2.3. Fluorescence In Situ Hybridization (FISH) Analysis

The commercial *MECOM (EVI1)* dual color, breakapart (BAP) DNA probe (#KI-10204) from Leica Biosystems/Kreatech (Buffalo Grove, IL, USA) and a home-brew tri-color *MECOM* probe were employed. The commercial probe is used routinely as a diagnostic test for all cases [10,43], whereas the home-brew probe is applied only in the cases with an atypical signal pattern obtained using the commercial probe to further confirm a positive status of *MECOM-R* in this study [51]. At least one metaphase *MECOM* FISH image with a decisive signal pattern and locations of all signals was captured in each case. In certain cases, mapback FISH was performed on a previously G-banded slide to better locate all FISH signals on the chromosome(s) involved. In addition, the Vysis LSI 5p15.1/*EGR1* probe set, *CEP7/7S522* probe set, *KMT2A(MLL)* BAP probe and *TP53/CEP17* probe set were applied in certain cases, as requested by clinicians. 

### 2.4. Morphological Examination

Peripheral blood (PB) and bone marrow (BM) aspirate smears were stained with Wright–Giemsa stain, and bone marrow core biopsy sections were stained with hematoxylin-eosin. The initial diagnostic BM biopsy carried out at the referral hospitals was sent to us for review (cases #1, 2, 5, 12, Table 1). The bone marrow core biopsy specimens were also stained with reticulin and trichrome for access to myelofibrosis, which was graded following the European Consensus on the grading of BM fibrosis [52].

### 2.5. Immunophenotyping by Flow Cytometry

Flow cytometric immunophenotypic analysis (PharmLyseTM, BD Biosciences, San Diego, CA, USA) was performed routinely on fresh bone marrow aspirate specimens as a part of the clinical work-up by following the standard procedures. All samples in this cohort were examined with antibody panels designed to assess leukemic cells, as reported previously [43], and the CD markers assessed include the following: CD2, surface and cytoplasmic CD3, CD4, CD5, CD7, CD13, CD14, CD15, CD19, CD25, CD33, CD34, CD36, CD38, CD41, CD45, CD56, CD64, CD117, CD123, HLA-DR, MPO and TdT. However, not all CD markers are assessed in all cases in this cohort.

### 2.6. Gene Mutation Profiling

A next-generation sequencing (NGS)-based analysis for the detection of somatic mutations in the coding sequences of 81 genes (81-gene panel) was routinely performed as part of the clinical work-up for all patients, as reported previously [53].

### 2.7. Statistical Analysis

Kaplan–Meier curves were employed to estimate the unadjusted overall survival (OS) duration. In this study, overall survival (OS) was calculated from the date of the first detection of pericentric inv(3) or the date of the initial diagnosis of the disease to death or the last follow-up. A student *t*-test was applied to perform all univariate analyses, and a chi-square (X^2^) test was utilized to compare the frequencies of different groups. All statistical analyses were conducted in GraphPad Prism 8, and statistical significance was considered if *p* < 0.05.

## 3. Results 

### 3.1. General Information and Clinicopathologic Characteristics

From 1 May 2009 to 30 September 2022, 283 myeloid neoplasm cases with *MECOM-R*, as confirmed by a FISH study, were identified in our institute; less than 50% of all cases exhibited a classic, paracentric inv(3)(q21q26.2) (n = 117, 41.3%) or t(3;3)(q21;q26.2) (n = 19, 6.7%), while the rest (n = 147, 51.9%) presented an atypical 3q26.2 aberration/*MECOM-R* through various mechanisms reported previously [10]. Out of this 3q26.2 aberration/*MECOM-R* cohort, we identified 17 cases (6%) with a pericentric inv(3), including 13 men and 4 women with a diagnosis of AML (n = 15), MDS (n = 1) or chronic myelomonocytic leukemia (CMML, n = 1) (Table 1) according to the 4th World Health Organization Classification [1]. Five patients had a prior history of other types of cancer and had been treated, including non-small cell lung cancer (case #11); prostate cancer (case #12), ovary cancer (case #13), diffuse large B-cell lymphoma (case #14) and Burkitt lymphoma (case #17). These five patients had been treated with chemotherapy and developed therapy-related AML (t-AML) or therapy-related MDS (t-MDS). Patient #10 had a history of CMML. Patients #3, #9 and #17 had a preceding history of MDS and transformed to AML. The median age was 67 years (range, 42 to 85) at the first detection of pericentric inv(3). All patients were treated with various chemotherapy regimens, as well as immunotherapy and targeted therapy in certain cases, such as nivolumab and ipilimumab in case #1; gemtuzumab in case #5; tegavivint (beta-catenin inhibitor) in case #6; PLX51107 (BRD4 inhibitor) in case #7; quizartinib in case #8; and gilteritinib in case #15. Two patients (cases #1 & #6) received a stem cell transplant (SCT); patient # 1 also received chimeric antigen receptor (CAR) NK cell therapy. By the endpoint of this study, 14 patients died and 3 were alive (1 partial remission and 2 progressive disease) (Table 1). The median overall survival was 4–5 months (range, 0 to 30 months) after the initial detection of pericentric inv(3) (Figure 1A and Table 1) or 14 months (range, 1 to 62 months) after the initial diagnosis of their myeloid neoplasm (Figure 1B and Table 1).

The PB and BM findings at the time of pericentric inv(3) detection are summarized in Table 2. All patients (100%) showed anemia, 15 patients showed thrombocytopenia and 11 (64.7%) patients showed pancytopenia. The median white blood cell (WBC) count was 6.9 × 10^9^/L (range, 0.5–96.5); the median hemoglobin level was 8.4 g/dL (range, 6.5–13.1); and the median platelet count was 29 × 10^9^/L (range, 7–225). The percentage of monocytes (normal: 2–7%) was increased in 15/17 patients (median 15%, range 0–52%). The median blast percentage was 8% (range, 0–76) in PB. BM cellularity was variable; the number of megakaryocytes was decreased in 11 (65%) cases, increased in 5 (29%) cases and adequate in 1 case. Monocytes were increased (normal <4%) in 11 (65%) cases. Morphologic dysplasia was found in all 15 cases who had enough cells to evaluate, 8 (53%) cases exhibited trilineage dysplasia. The most characteristic finding was megakaryocytes, which frequently were small and hypolobated (Figure 2A,B). Seven of the thirteen cases evaluated had bone marrow fibrosis, and seven showed mild reticulin fibrosis (MF-1).

### 3.2. Immunophenotypic Features

All cases in this cohort showed a myeloid immunophenotype, CD13+/CD33+/CD34+/CD117+. All cases expressed CD123+, and all cases except for one were positive for HLA-DR. CD7 was positive in 12 of 17 cases, myeloperoxidase was positive in 6 of 14 cases assessed, CD64 was positive in 3 of 17, CD19 was positive in 2/17 and CD5 was positive in 1/17 cases. All cases were negative for CD14. The numbers of tested and positive cases for each marker/antigen are summarized in Appendix A.

We also compared the antigen expression between cases with −7/7q abnormalities (n = 12) and cases without −7/7q abnormalities (n = 5); no statistically significant differences were observed (see Appendix A).

### 3.3. Cytogenetic Characteristics

The involved band levels at 3p were determined as follows: 1 at 3p13 (case #1), 2 at 3p21 (cases #2–3), 11 at 3p23 (cases #4–14) and 3 at 3p25 (cases #15–17), (Table 3). The inv(3)(p23q26.2) and inv(3)(p25q26.2) are considered as subtle or even cryptic abnormalities by conventional cytogenetics (Figure 3). Nine (53%) neoplasms had a complex karyotype, six (35.3%) cases had additional chromosomal aberration and two had pericentric inv(3)s as the sole chromosomal aberration. Twelve (70.6%) cases simultaneously exhibited 7q aberrations including monosomy 7 (−7, n = 10), 7q deletion (7q-, n = 1) or ring chromosome 7 or r(7) (n = 1).

Sixteen patients had karyotyping and/or *MECOM* FISH information at other hospitals before they were referred to our cancer center, but pericentric inv(3) was identified in only one case (case #2) at a referral hospital. *MECOM-R* was reported by referral hospitals in three cases (cases #1, #2 and #12), and the status of pericentric inv(3)/*MECOM-R* was missed in the remaining 12 cases at the referral hospitals. We retrospectively checked the count sheet used by our technologists for chromosomal analyses, though an abnormal chromosome 3 or der(3) was identified by our technologists in 7 cases, and a pericentric inv(3) abnormality was initially missed in 13 cases. The median interval between the detection of pericentric inv(3)s/*MECOM-R* and the initial diagnosis of diseases was 7 months (range, 0 to 50) in this cohort (Table 3).

All cases were *MECOM* BAP FISH-positive, and a split signal pattern with *3ʹMECOM* signal (green) locating on 3p and *5ʹMECOM* signal (red) locating on 3q of the affected chromosome 3 documented by metaphase FISH and/or mapback FISH was observed in 13 cases. Four cases (cases #2, #8, #14 and #16) showed an atypical split signal pattern [10] with a fusion signal (yellow) located on 3p and *5ʹMECOM* signal (red) located on 3q of the affected chromosome 3 (e.g., cases #2 and #16 in Figure 3). The latter four cases were further confirmed with a home-brew, tri-color MECOM BAP FISH for their *MECOM-R* status (Appendix A) [51].

### 3.4. Gene Mutation Profiles

A total of 52 mutation events affecting 23 genes were identified in this cohort, with an average of 3 mutations per case. All cases exhibit mutation(s) affecting one to eight genes simultaneously (one gene, n = 3; two genes, n = 7; three genes, n = 1; four genes, n = 3; five genes, n = 1; seven genes, n = 1 and eight genes, n = 1) (Figure 4). *NRAS* showed the highest frequency of mutation (n = 6), followed by *RUNX1* (n = 5), *FLT3* (n = 4), *ASXL1*, *PTPN11*, *SF3B1* and *TP53* (n = 3). These 7 genes accounted for 27 of the 52 (53%) mutation events. Sixteen other genes were mutated at a low frequency (n = 2 or n = 1) (Figure 4).

## 4. Discussion

The first myeloid neoplasm associated with a pericentric inv(3) was reported by Grigg et al. in 1993 [54]. The patient presented with inv(3)(p25q27) and was categorized as one of the miscellaneous 3q abnormalities in a cohort of 24 cases with various 3q abnormalities, mostly classic inv(3)/t(3;3)(q21;q26.2). Three additional cases were reported by Heimann et al. [55], Shi et al. [56] and Smith et al. [57], respectively. However, the status of *MECOM* rearrangement was not clarified and remained unknown in all of these four cases. Poppe et al. reported the first pericentric inv(3) case with *MECOM-R*, as confirmed by FISH [58]. The patient was reported to have an inv(3)(p12q26), an apparent abnormality of chromosome 3. Since then, several groups reported various forms of pericentric inv(3), and most of these cases had *MECOM-R*, as confirmed by FISH (Table 4) [3,16,59,60,61,62,63]. Therefore, to the best of our knowledge, a total of 26 cases with pericentric inv(3) have been reported, including an inv(3)(p21.3q26.2) case reported by Dias et al. [62], which is considered a constitutional abnormality transmitted from donor to recipient but without causing any diseases.

Here, we add 17 cases for a total of 33 cases with pericentric inv(3)/*MECOM-R* [3,16,58,61,63]. However, the band levels of 3p involvement varied very much among these cases, e.g., p25 (n = 5), p24 (n = 11), p23 (n = 12), p21 (n = 3) and p13 (n = 2) (Table 3 and Table 4). Locus 3p24 and 3p23 involvement account for over two-thirds of these cases, and these two adjacent bands can barely be distinguished from each other if the resolution of a metaphase cell is <400-band levels. Interestingly, most cases with 3p25, 3p24 and 3p23 are considered as subtle or even cryptic by chromosome analysis, as described by others [16,61]. Without *MECOM* BAP FISH testing with split signals located on both the *p* and q arms of the affected chromosome 3 simultaneously (Figure 3), the pericentric inv(3)s would not have been recognized/confirmed in these cases. In fact, as mentioned previously, almost all of these pericentric inv(3) abnormalities were missed initially by chromosomal analyses in this cohort, and some of these abnormalities were missed multiple times. The add-on *MECOM* FISH by clinicians and/or the reflex FISH test due to the presence of −7/7q- helped to discover the *MECOM-R* as well as the pericentric inv(3)s in this study. A further analysis of the karyotype results of all cases with subtle/cryptic pericentric inv(3), including 13 cases in this study and 15 cases in the literature, revealed a high frequency of −7 and 7q- which was about 75%, higher than that in cases with classic inv(3)/t(3;3) or other 3q26.2/*MECOM-R* abnormalities (<22%) [3,9,10,11]. More interestingly, −7 or 7q- was the only additional chromosomal aberration in 14 cases. Therefore, −7 and 7q- including r(7) can be considered as an indicator for performing MECOM FISH to confirm/exclude a pericentric inv(3)/*MECOM-R* status in these cases.

The recognition of pericentric inv(3) with MECOM-R is clinically important for neoplasm classification and risk stratification in affected patients [1,3,9,11]. The detection of pericentric inv(3) may cause the cytogenetics risk group to change from intermediate-risk to high-risk in cases #2 and #8 and change the diagnosis from MDS to AML in cases #10 and #14 under the 5th edition of the WHO Classification of Myeloid Neoplasms, where AML with MECOM-R is a subentity of AML regardless of blast counts [6,27]. Lastly, although there is no targeted therapy specifically against MECOM-R currently, several studies have shown that certain targets and/or existing therapeutical drugs may be promising for an effective treatment of MECOM-R AML. For example, Fenouille et al. [64] reported that an ATP-buffering mitochondrial creatine kinase (CKMT1A) is tightly associated with MECOM/EVI1 expression in AML, and the administration of cyclocreatine, a CKMT1A inhibitor, can dramatically reduce the viability of MECOM/EVI1-expressing AML cells. Recently, poly(ADP-ribose) polymerase 1 (PARP1) inhibitors have also been identified as a potential new therapy for MECOM-R AML patients [40]. Since there are several FDA-approved PARP inhibitors available as well as ongoing clinical trials involving treating AML/MDS with PARP inhibitors [65,66], clinical trials involving treating MECOM-R AML patients with PARP inhibitors may be available soon. Therefore, the immediate identification of cryptic MECOM-R-associated chromosomal abnormalities is crucial for the recruitment of qualified patients for clinical trials once targeted therapies become fully available. FISH testing, especially that which utilizes a breakapart (BAP) FISH probe, is currently the main method applied clinically for the exclusion/confirmation of MECOM-R. However, a BAP FISH test will not provide information on the partner gene for MECOM-R, if any exists, and one BAP FISH test may not always provide a definite conclusion in some cases due to the complexities of 3q26.2 abnormalities and the wide range of breakpoints involving MECOM as well as its flanking regions [9,10,51]. NGS-based whole genome structural variant (SV) profiling methods, such as optical genome mapping (OGM) [67], whole exome sequencing (WES) [36,68,69], whole genome sequencing (WGS) [7,30,36,69] and/or whole transcriptome sequencing (WTS) [30,36], have been successfully applied in the diagnosis of AML/MDS cases with MECOM-R through cryptic 3q26.2 aberrations for chromosomal analysis. Fusion genes involving MECOM have been identified in all these cases. We would like to point out that we have not investigated any of these cases with pericentric inv(3)/MECOM-R with one of these NGS-based whole genome structural variant (SV) profiling methods yet, but they are all worthy of being further explored. 

Patients with pericentric inv(3) shared similar clinicopathologic features with the classic inv(3)(q21q26.2): anemia, morphologic dysplasia (including characteristic dysmorphic megakaryocytes) and inferior outcomes. On the other hand, patients with pericentric inv(3) more frequently showed increased monocytes in PB and BM, thrombocytopenia and decreased megakaryocytes compared to patients with classic inv(3)/t(3;3), which could largely be due to the difference in the partner genes on 3p other than *GATA1* on 3q21. An average of three mutation events per case has been observed in our cohort with pericentric inv(3)/*MECOM-R*, which is slightly higher than the average of two mutation events per case observed in AML with classic inv(3)/t(3;3) [11,61,63,68]. *NRAS* mutations showed the highest frequency, about 35%, followed by *RUNX1* and *FLT3* mutations, similar to the mutation profiles of the pericentric inv(3) cases reported by other groups [61,63]. It is relevant to point out that *SF3B1* mutations are common in classic inv(3)/t(3;3) cases [61,67]. In contrast, only three cases with pericentric inv(3) in this cohort had an *SF3B1* mutation.

In summary, we report 17 patients with pericentric inv(3)/*MECOM-R* with different breakpoints on 3p. Along with the cases reported in the literature, inv(3)(p25q26.2), inv(3)(p24q26.2) and inv(3)(p23q26.2) can be subtle or even cryptic and thus be easily overlooked by conventional chromosomal analysis. About 75% of the cases in this study have monosomy 7 or deletions of 7q that can serve as an indicator to reflex MECOM FISH to further exclude/confirm the subtle/cryptic aberrations. The immediate recognition of pericentric inv(3)/*MECOM-R* is associated with the diagnosis and risk stratification of certain patients.

## 5. Conclusions

The chromosomal aberrations associated with *MECOM-R* can be subtle or even cryptic and thus be easily overlooked by karyotyping only, including the majority of cases with pericentric inv(3)s in this study. The early detection and confirmation of 3q26.2 aberrations/*MECOM-R* are relevant for the diagnosis, risk stratification and clinical management of the affected patients. Our study indicates that cases with 7q-/−7 are worthy of being further investigated for a potential 3q26.2 aberration/*MECOM-R*—for example, performing a reflex MECOM FISH test.

## Figures and Tables

**Figure 1 cancers-15-00458-f001:**
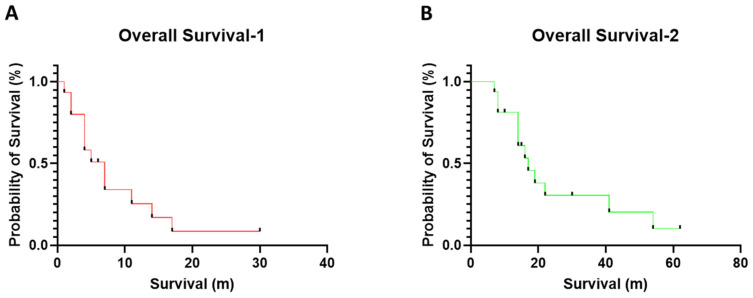
The overall survivals in this cohort with 3q26 aberration/*MECOM-R* myeloid neoplasms. (**A**) The overall survival was calculated from the time of the initial detection of pericentric inv(3)s; (**B**) the median overall survival was calculated from the initial diagnosis of myeloid neoplasm.

**Figure 2 cancers-15-00458-f002:**
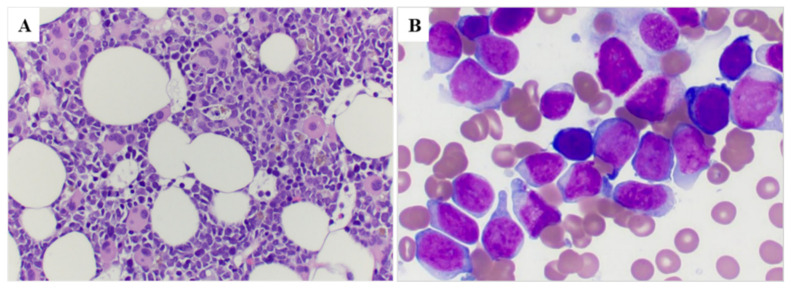
Bone marrow morphology in a representative case (case #8). (**A**) Bone marrow core biopsy shows a hypercellular marrow with increased immature cells as well as many dysplastic megakaryocytes with small hypolobated or separated nuclear lobes (100×); (**B**) Bone marrow smear shows increased blasts that are large with dispersed chromatin and distinct nucleoli (500×).

**Figure 3 cancers-15-00458-f003:**
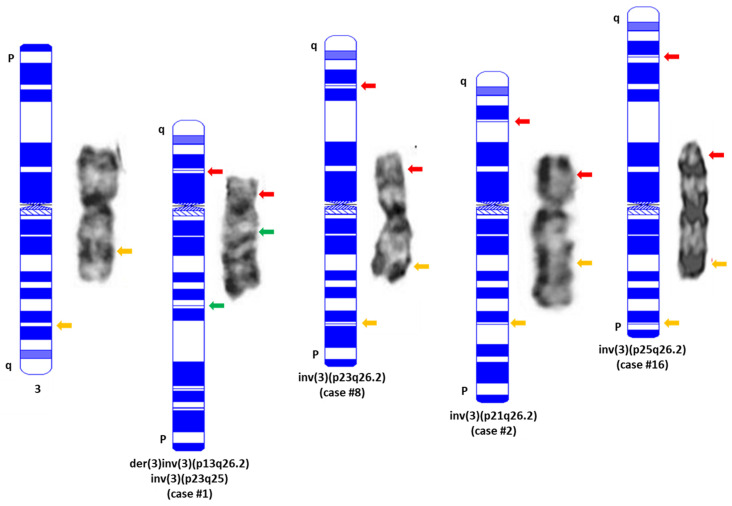
Representative images of normal chromosome 3 and pericentric inv(3)s detected in this study. A karyogram drawn using online software (CyDAS, http://www.cydas.org/OnlineAnalysis/ accessed on 10 November 2022) is placed on the left side of each chromosome with an indication of the sites and colors of MECOM BAP FISH using the two-color commercial probe in this study. Yellow: intact MECOM signal; Red: 5′MECOM; Green: 3′MECOM. It is necessary to point out that all the pericentric inv(3)s listed in this figure are intentionally exhibited in a direction from q (**top**) to *p* (**bottom**) after inversion so that their centromeres remain as the normal chromosome 3.

**Figure 4 cancers-15-00458-f004:**
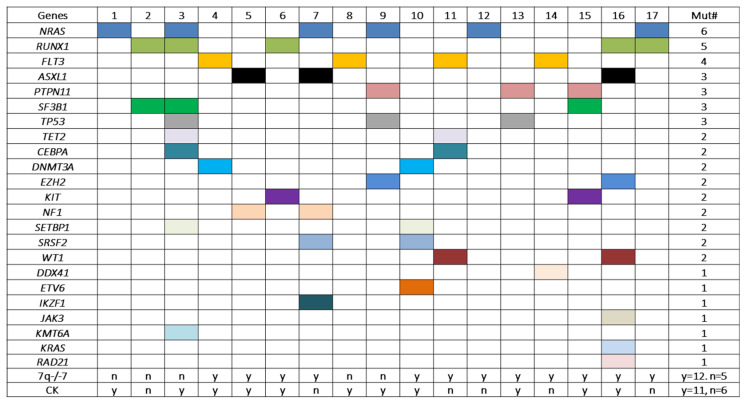
Gene mutations detected in patients with pericentric inv(3)s in this cohort. Each colored square represents a mutation event. All cases carried mutation(s) affecting at least one gene. Cases #16 and #3 had mutations involving seven and eight genes simultaneously. The first six genes (*NRAS, RUNX1, FLT3, ASXL1, PTPN11, SF3B1* and *TP53*) from the top showed frequencies of mutation of 3/17 to 6/17. However, no statistically significant association was found between the mutation frequencies and the status of −7/7q- and/or complex karyotype. Mut: mutation; CK: complex karyotype; n: no; y: yes.

**Table 1 cancers-15-00458-t001:** Demographic and Clinical Information.

Case	Age (y)/Sex	Diagnosis *	Treatment	Outcome	OS-1 (m)	OS-2 (m)
1	55/M	AML	Multi-lines of chemotherapy, cord blood SCT Ipilimumab, Nivolumab; NK-CAR	D	41	41
2	80/M	AML	Multi-lines of chemotherapy	D	35	36
3	72/F	AML-MRC	Multi-lines of chemotherapy	PD	2	62
4	70/M	AML	Multi-lines of chemotherapy	D	2	7
5	61/M	AML	Multi-lines of chemotherapy, gemtuzumab	D	5	22
6	54/M	AML	Multi-lines of chemotherapy; SCT; tegavivint (beta-catenin inhibitor)	D	11	14
7	70/M	AML	Multi-lines of chemotherapy; PLX51107 (BRD4 inhibitor)	D	7	14
8	85/F	AML-MRC	Multi-lines of chemotherapy, quizartinib	D	14	17
9	69/F	AML-MRC	Multi-lines of chemotherapy	D	4	54
10	81/M	CMML	Multi-lines of chemotherapy; then maintained with transfusion	PR	6	10
11	67/M	t-AML	Multi-lines of chemotherapy	D	4	19
12	78/M	t-AML	Multi-lines of chemotherapy	D	7	16
13	45/F	t-AML	Ara-C	D	1	1
14	76/M	t-MDS	Not treated for MDS	D	0	8
15	53/M	AML	Multi-lines of chemotherapy, gilteritinib	D	2	8
16	66/M	AML	Multi-lines of chemotherapy	D	4	14
17	42/M	t-AML	Multi-lines of chemotherapy	PD	1	10

AML: acute myeloid leukemia; AML-MRC: acute myeloid leukemia (AML) with myelodysplasia-related changes; MDS: myelodysplastic syndromes; CMML: chronic myelomonocytic leukemia; D: death; F: female; M: male; m: month; y: year; PD: progressive disease; PR: partial remission; t-: therapy-related; OS-1: overall survival calculated from the date of detection of pericentric inv(3); OS-2: overall survival calculated from the initial diagnosis of disease; CAR: chimeric antigen receptor; SCT: stem cell transplant; Ara-C: cytarabine. * Following 2017 WHO classification.

**Table 2 cancers-15-00458-t002:** Peripheral blood and bone marrow findings.

Case	Peripheral Blood Findings					Bone Marrow Findings					
	WBC (10^9^/L)	Hgb (g/dL)	Plt (10^9^/L)	Blasts (%)	Mono (%)	Cellularity (%)	Blasts (%)	Mono (%)	Meg	Dysplasia	MF
1	0.7	6.6	236	22	16	90	90	0	Dec	Meg, E	NA
2	3.7	7.6	104	3	52	55	42	14	Inc	Meg, G, E	MF-0
3	1	10	40	3	9	50	27	6	Inc	Meg, G, E	MF-0
4	1.3	10.9	29	30	12	75	54	2	Dec	NA *	MF-0
5	6.65	7.9	44	0	43	80	27	9	Dec	G, E	NA
6	1.4	9.5	46	72	0	80	46	11	Inc	Meg, G	MF-1
7	7.3	9.6	21	20	18	90	57	1	Dec	Meg, G, E	MF-1
8	15.4	7.7	20	4	22	80	30	9	Dec	Meg, G, E	MF-1
9	1.5	8.9	15	8	14	20	26	21	Dec	Meg, G	MF-1
10	6.7	8	225	0	26	70	4	14	Inc	Meg, G, E	MF-0
11	46.5	6.5	11	42	47	95	73	16	Dec	NA *	MF-1
12	3.7	10.7	44	17	22	80	10	1	Dec	G, E	NA
13	96	8	11	3	15	65	33	9	Dec	Meg, G, E	MF-0
14	20.3	9.3	121	0	15	50	1	4	Inc	Meg, G, E	NA
15	96.5	8.6	116	76	4	100	56	8	Dec	Meg, G, E	MF-1
16	2.2	6.8	29	10	8	60	31	5	Dec	Meg, E	NA
17	9.1	8	19	17	20	60	21	2	Ade	E	MF-1

Ade: adequate; Dec: decreased in number; E: erythrocytes; G: granulocytes; Hgb: hemoglobin; Meg: megakaryocytes; MF: myelofibrosis; Plt: platelets; Meg: megakaryocytes; WBC: white blood cells. * too few cells to evaluate.

**Table 3 cancers-15-00458-t003:** Cytogenetic features of cases with pericentric inv(3)/*MECOM-R* in this cohort.

Case	Final Karyotype	Pericentric Inv(3)	7q-/−7	*MECOM* FISH	Outside Reports	Interval (m)
1	46~47,XY,der(3)inv(3)(p13q26.2)inv(3)(p23q25),add(4)(q21),−6,del(9)(q21),der(10)t(1;10)(q12;p12),−20,+2mar[cp4]//46,XX[16]	inv(3)(p13q26.2)	no	pos	der(3), *MECOM*-R by FISH	24
2	46,XY,inv(3)(p25q26.2)[18]/46,XY[2]	inv(3)(p21q26.2)	no	pos *	inv(3), *MECOM*-R by FISH	0
3	48,XX,del(5)(q22q35),+8, +8[14]/44,XX,+ 1,der(1;14)(q10;q10),inv(3)(p21q26.2),del(5)(q22q35),add(14)(p11.2),−16,−20[6]	inv(3)(p21q26.2)	no	pos	5q-	0
4	45,XY,inv(3)(p23q26.2),-7,add(8)(q24.1),del(20)(q11.2q13.1)[20]	inv(3)(p23q26.2)	yes	pos	t(3;8), −7, del20q	5
5	45,XY,inv(3)(p23q26.2),−7,t(17;21)(q11.2;q22)[15]/45,idem,der(22)t(1;22)(q21;p12)[5]	inv(3)(p23q26.2)	yes	pos	−7	16
6	46,XY,inv(3)(p23q26),−7,+21[20]	inv(3)(p23q26.2)	yes	pos	−7	2
7	45,XY,inv(3)(p23q26.2),−7[8]/46,XY[12]	inv(3)(p23q26.2)	yes	pos	−7	7
8	46,XX,inv(3)(p23q26.2)[18]/45,idem,t(4;5)(q21;p15.1),−21[1]/47,XX,add(5)(p15.3),+13[1]	inv(3)(p23q26.2)	no	pos *	+13	2
9	46,XX,+1,der(1;16)(q10;p10),inv(3)(p23q26.2),del(5)(q13q33),add(7)(p13)[cp5]/71,XXX,+1,der(1;16)(q10;p10),inv(3)(p23q26.2)x2,del(5)(q13q33),+19,+21[3]/66~71,XXX,+1,der(1;16)(q10;p10),inv(3)(p23q26.2)x2,del(5)(q13q33),+19,+21,+22[cp12]	inv(3)(p23q26.2)	no	pos	−5, + 8, der(16)t(1;16)(q21;q12)	50
10	45,XY,inv(3)(p23q26.2),−7,del(11)(q21)[5]/46,XY[15]	inv(3)(p23q26.2)	yes	pos	−7, del11q	12
11	46,XY,inv(3)(p23q26.2),r(7)[20]	inv(3)(p23q26.2)	yes	pos	−7	4
12	45,XY,inv(3)(p23q26.2),−7[19]/46,XY[1]	inv(3)(p23q26.2)	yes	pos	−7, *MECOM*-R by FISH	8
13	44~45,XX,add(1)(p13),add(2)(q31),inv(3)(p23q26.2),del(4)(q28),−5,−7,del(12)(p13),+mar[cp20]	inv(3)(p23q26.2)	yes	pos	n/a	0
14	46,XY,inv(3)(p23q26.2),del(7)(q22q34)[10]/46,XY[10]	inv(3)(p23q26.2)	yes	pos *	add7q	8
15	45,XY,inv(3)(p25q26.2),−7[19]/40,idem,−4,−6,−10,add(11)(q22),add(17)(p13),−19,−21[1]	inv(3)(p25q26.2)	yes	pos	n/a	5
16	45,XY,inv(3)(p25q26.2),−7[2]/47,idem,+8,+21[18]	inv(3)(p25q26.2)	yes	pos *	−7	9
17	45,XX,inv(3)(p25q26.2),−7[20]	inv(3)(p25q26.2)	yes	pos	−7	10

pos: positive; n/a: not available; m: months. Outside reports: chromosomal analysis and/or FISH results provided by referral hospitals. These tests were performed and obtained before the patients were referred to our hospital. Interval: from the initial diagnosis of myeloid neoplasm to the detection of inv(3) and/or *MECOM*-*R*. * Atypical split signal pattern (split red signal and two fusion signals, 1R2F using the commercial *MECOM* FISH probe).

**Table 4 cancers-15-00458-t004:** A summary of cases with pericentric inv(3)s reported in the literature.

Year of Publication	Ref#	PMID	Pericentric inv(3)s	# of Cases Reported	*#* of *MECOMre+* by FISH	*#* of *MECOMre−* or Unknown
1993	[54]	8435325	inv(3)(p25q27)	1	0	1
1994	[55]	8194049	inv(3)(p21q27)	1	0	1
1996	[56]	8976389	inv(3)(p21q26)	1	0	1
2001	[57]	11672770	inv(3)(p21q26)	1	0	1
2006	[58]	16342172	inv(3)(p12q26)	1	1	0
2007	[59]	17976519	inv(3)(p13q26)(n = 2)	1	0	1
2010	[3]	20660833	inv(3)(p21q26), inv(3)(p25q26) (n = 2), inv(3)(p13q26), inv(3)(p21q27~29), inv(3)(p21q27)	6	3	3
2012	[60]	22403058	inv(3)(p23q26)	1	0	1
2012 *	[16]	22887804	inv(3)(p24q26) (n = 10)	10 *	10 *	0
2017	[62]	28549770	inv(3)(p21.3q26.2) ** transmitted from donor to recipient	1	0	1
2020 *	[61]	32189545	inv(3)(p24q26) (n = 9)	9 *	9 *	0
2022	[63]	34668265	inv(3)(p23q26.2), inv(3)(p24q26.2)	2	2	0

* Reports from the same group. To avoid a potential duplicate of cases, the larger number is applied for statistical analysis. ** Constitutional abnormality.

## Data Availability

The data presented in this study are available in this article (and Appendix A).

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
