# Peer review of "3q26.2/MECOM Rearrangements by Pericentric Inv(3): Diagnostic Challenges and Clinicopathologic Features"

_cancers, 2023, doi:10.3390/cancers15020458_

Round 1

Reviewer 1 Report

This is impactful work and a well designed and executed study.

Some comments/critiques below:

1) In the abstract, please include the M/F and age stats (median, range).

2) In the abstract, consider changing "characterized" to some unique features. 

 3) can you compare survival to a cohort of "classic" AML with MECOM-R cases?

4) You wrote in the discussion- "It’s necessary to point out that SF3B1 mutations are common in classic inv(3)/t(3;3) cases". Please use formal English- It's should be something more like "It is relevant" .

5) There are more cases of pericentric Inv(3) to add in Table 4 from the study: Summerer, I., et al., Prognosis of MECOM (EVI1)-rearranged MDS and AML patients rather depends on accompanying molecular mutations than on blast count. Leuk Lymphoma, 2020. 61(7): p. 1756-1759. Please add those in as well.

6) in the abstract- consider adding in a sentence about molecular features. for instance- the lower frequency of SF3B1 mutations compared to classic rearranged cases, and also NRAS,FLT3,RUNX1 mutations.

Author Response

Thank you for reviewing our manuscript. Please find below our point-by-point reply to your comments/critiques.

1. In the abstract, please include the M/F and age stats (median, range).

Reply: This is a very good suggestion. These information have been added. Please see Lines #35 and #36 in the revised manuscript.

2. In the abstract, consider changing "characterized" to some unique features. 

Reply: Changes are made. Please see Line #46 in the revised manuscript. Thank you!

3. can you compare survival to a cohort of "classic" AML with MECOM-R cases?

Reply: This is a very important suggestion. A comparison of overall survival between “classic” and pericentric inv(3) groups has not been included in this manuscript due to several facts: the cases with “classic” inv(3) is overnumbered the pericentric inv(3) group ( 117 vs. 17, please see Lines #177 to #182 in the revised manuscript); many “classic” cases had lost follow-up; many cases have been diagnosed/treated outside, and the detailed information in this regard was missing…

4. You wrote in the discussion- "It’s necessary to point out that SF3B1 mutations are common in classic inv(3)/t(3;3) cases". Please use formal English- It's should be something more like "It is relevant" .

Reply: Changes are made (please see Line #364 in the revised manuscript). Thank you!

5. There are more cases of pericentric Inv(3) to add in Table 4 from the study: Summerer, I., et al., Prognosis of MECOM (EVI1)-rearranged MDS and AML patients rather depends on accompanying molecular mutations than on blast count. Leuk Lymphoma, 2020. 61(7): p. 1756-1759. Please add those in as well.

Reply: This study from Summerer et al has been already included in the Table 4 (reference # 61).

6. in the abstract- consider adding in a sentence about molecular features. for instance- the lower frequency of SF3B1 mutations compared to classic rearranged cases, and also NRAS,FLT3,RUNX1 mutations.

Reply: Changes are made (please see Line #42 and #44 in the revised manuscript). Thank you!

Reviewer 2 Report

This is a detailed report of 17 patients with myeloid neoplasms (predominantly AML) with MECOM rearrangements associated with pericentric inv(3) from a single (large) institution genetic database. The authors have made a detailed description of the chromosome breakpoints and clinico-pathological disease associations which adds considerably to the existing knowledge of this leukaemia-associated abnormality. It is written in good English throughout with relatively few typographical errors which should be easy to edit with further proof-reading.

A few relatively minor specific points / questions:

1. It would have been helpful to give an idea of the number of samples on the institutional database from the 2009-2022 period that these 17 cases were accrued from. After reading the article I still didn't have a clear feel for the perceived frequency of these pericentric inv(3) rearrangements or how this is felt to compare with the incidence of the conventional inv(3) abnormality. There will perhaps be some limitations here due the the MD Anderson database containing a higher proportion of already relapsed / refractory cases then would be found in a less specialised centre.

2. The majority of these analyses (15 out of 17 I think) were performed on samples taken at times other than first AML diagnosis. For some of these samples (table 3) an earlier cytogenetic report is quoted. It remains unclear whether it is felt that the pericentric inv(3) is felt to be an abnormality that is acquired at the point of disease progression / relapse or it is just that the abnormality isn't generally spotted at first diagnosis. Was it possible to perform a lookback at any archived samples from earlier points in any of the patient journeys?

3. Following on from point 2. Table 3 would have benefitted from a greater explanation of what the columns headed 'outside reports' and 'interval (m)' are referring to. Were the outside reports all of samples taken at first diagnosis or were some taken at later points, but prior to referral to MD Anderson?

4. I appreciate that numbers are limited, but was it possible to see any prognostic difference between cases with associated chromosome 7 abnormalities, and those without?

Author Response

Thank you for reviewing our manuscript. Please find below our point-by-point reply to your comments/critiques.

  1. It would have been helpful to give an idea of the number of samples on the institutional database from the 2009-2022 period that these 17 cases were accrued from. After reading the article I still didn't have a clear feel for the perceived frequency of these pericentric inv(3) rearrangements or how this is felt to compare with the incidence of the conventional inv(3) abnormality. There will perhaps be some limitations here due the the MD Anderson database containing a higher proportion of already relapsed / refractory cases then would be found in a less specialised centre.

Reply: Thank you for your good suggestion. We have added the information as you have suggested. Please see Lines #177 to #182 in the revised manuscript.

  1. The majority of these analyses (15 out of 17 I think) were performed on samples taken at times other than first AML diagnosis. For some of these samples (table 3) an earlier cytogenetic report is quoted. It remains unclear whether it is felt that the pericentric inv(3) is felt to be an abnormality that is acquired at the point of disease progression / relapse or it is just that the abnormality isn't generally spotted at first diagnosis. Was it possible to perform a lookback at any archived samples from earlier points in any of the patient journeys?

Reply: You are absolutely right. The initial detection/determination of a pericentric inv(3) was not made at times of the first AML diagnosis in 15 out of 17 cases in this study. However, the detection of -7/7q- as well as positive MECOM FISH results reported outside in some cases indicate that a pericentric inv(3) might exist but was overlooked at the referral hospitals before these patients were first seen in our hospital, although we are uncertain whether the pericentric inv(3) did exist at the initial AML diagnosis or occur during disease progression in each case.  Unfortunately, we are unable to obtain any archived samples from referral hospitals and/or commercial diagnostic companies to perform a lookback.

  1. Following on from point 2. Table 3 would have benefitted from a greater explanation of what the columns headed 'outside reports' and 'interval (m)' are referring to. Were the outside reports all of samples taken at first diagnosis or were some taken at later points, but prior to referral to MD Anderson?

Reply: Thank you for pointing this out. A note to explain the outside reports and interval have been added. Please see Lines #248 to #250 in the revised manuscript. From the information provided by referral hospitals, their test results were obtained from a specimen after a diagnosis of myeloid neoplasm had been established.

  1. I appreciate that numbers are limited, but was it possible to see any prognostic difference between cases with associated chromosome 7 abnormalities, and those without?

Reply. Thank you for bringing up this interesting point. Actually, we compared the overall survival between cases with vs. without 7q-/-7 in this cohort, and our result showed that median survivals after initial detection of pericentric inv(3) in these two groups did not show statistical significance; however, the median overall survival after the initial diagnosis of their myeloid neoplasm is significantly higher in the group with 7q-/-7 (41 months) than that without 7q-/-7 (14 months) (p<0.01). However, this might be caused by the bias of cases numbers in two groups (12 vs. 5) and many other facts, such as stem cell transplant, etc. Therefore, we decided not to include this information in the manuscript so that the readers will not be confused.